# Evolution of Sow Productivity and Evaluation Parameters: Spanish Farms as a Benchmark

**DOI:** 10.3390/vetsci11120626

**Published:** 2024-12-06

**Authors:** Santos Sanz-Fernández, Pablo Rodríguez-Hernández, Cipriano Díaz-Gaona, Llibertat Tusell, Raquel Quintanilla, Vicente Rodríguez-Estévez

**Affiliations:** 1Departamento de Producción Animal, UIC Zoonosis y Enfermedades Emergentes ENZOEM, Facultad de Veterinaria, Campus de Excelencia Internacional Agroalimentario (ceiA3), Universidad de Córdoba, Campus de Rabanales, 14071 Córdoba, Spain; v22safes@uco.es (S.S.-F.); v22rohep@uco.es (P.R.-H.); pa2digac@uco.es (C.D.-G.); 2Animal Breeding and Genetics Program, IRTA, Torre Marimon, 08140 Barcelona, Spain; llibertat.tusell@irta.cat (L.T.); raquel.quintanilla@irta.cat (R.Q.)

**Keywords:** sow efficiency, farm sustainability, reference parameters, pig production, piglet survival, animal welfare

## Abstract

Pig production plays a crucial role in the global meat supply, and Spain is emerging as one of the leading producers. This study explores the evolution of sow productivity and farm efficiency on a global scale, with a particular focus on Spain over the last decade, considering both commercial and Iberian sow farms. By analyzing key factors such as the number of piglets weaned per sow, prolificacy, and mortality rates, this study identifies trends and challenges that affect farm sustainability and animal welfare, highlighting the need for new management strategies and broader benchmarking parameters. The study suggests the use of innovative indicators that go beyond traditional productivity measures, incorporating factors such as piglet survival, sow longevity, animal welfare, and environmental sustainability. These findings provide valuable insights for producers to improve farm efficiency, contributing to a more sustainable and responsible swine industry.

## 1. Introduction

Pig production plays a significant role in terms of global meat supplies, accounting for 34% of the world’s meat consumption [1] even though a considerable portion of the population does not consume pork for religious reasons or due to being generally reluctant to consume meat [2]. Alongside pork consumption, global pork production has increased by approximately 130% over the past 60 years from 1962 to 2022 [3].

### 1.1. The Global and European Union Pork Sector

Currently, the leading pork producers in the world are China (46.8%), the European Union 27 (13.7%), and the United States (7.6%), which together account for more than 68% of global pork production (Figure 1) [3]. This context means that any changes in productivity or census variations in the main producing countries have a direct impact on global pork production and the pork market. In this regard, the significant reduction in pork production caused by the outbreak of African Swine Fever (ASF) in East Asia [4] resulted in a loss of approximately 27.9 million tons from 2018 to 2021 in China. Pork prices in China more than doubled despite a surge of pork exports from the European Union, United States, Canada, Brazil, and other countries. Pork exports to China prompted smaller price increases in exporting countries’ pork markets [5].

The increase in the number of pigs globally produced is largely due to an improved production efficiency. In terms of efficiency, there are significant differences among the leading pork-producing countries: the European Union (EU) and the United States (USA) stand out, producing 40% and 38% more pork with the same number of pigs compared to China, respectively [4]. According to data from the Web of Science (WOS), the number of published articles related to pig production has increased exponentially in recent years from 219 articles in 1963 to figures nearing 4500 articles published annually since the beginning of the current decade (Figure 2). Currently, the number of scientific papers published on this type of production exceeds 110,000 articles [6]. Similarly, rising trends are also observed in research about sow productivity, with a significant increase in published studies in recent years (Figure 2) [6]. These figures demonstrate not only the great development in terms of technology and specialization that this sector has experienced over time to achieve these numbers but also the extensive research efforts required for these advances.

In this context, and despite the ongoing consequences of ASF, global pork production is expected to continue growing, with an estimated 17% increase (18 million tons) by 2031 compared to the 2019–2021 period [7]. Therefore, ongoing research, as well as the study and development of new efficiency evaluation parameters for pig production, is still necessary.

Since the 1990s, the EU has experienced a steady increase in pork production. However, this growth has been limited in recent years due to outbreaks of infectious diseases (e.g., ASF, PRRSV), coupled with stricter environmental laws and animal welfare regulations, which explain the slowdown in pork production growth in Europe [8]. Thus, after 2021, EU pork production dropped from 23.39 million tons to 20.64 million tons in 2023 [9]. Within the EU, the countries with the highest pork production levels are Spain, Germany, and France, which, together, account for 54% of total European production [9]. Other countries, such as Denmark, stand out for their high productivity levels, having the lowest production costs in Europe [10] due to the high efficiency of their sows [11] and their great disease control and elimination plans, especially for PRRSV. The evolution of the pig population in the EU has also been uneven (Figure 3); while Spain increased its global pig census by 35.13% over the last 10 years, Germany, in second place, experienced a 25% decrease in its census, as did other leading European producers such as France (−12%), Denmark (−8%), The Netherlands (−13%), and Poland (−11%) [12]. Part of that decrease in production is due to the recent ASF outbreaks (after the introduction in each country), although it is not possible to specify. This has allowed Spain to become the leading pig-producing country, followed by Germany, in the EU today.

### 1.2. Situation of the Pig Sector in Spain Compared to Other Producers

In general, Spain ranks as the third-largest pork producer in the world (Table 1), with a total of 52,989,958 animals slaughtered in 2023 and approximately 4.9 million tons of pork being produced, representing about 24% of the EU’s production [13]. To explain this scenario of pork consumption and pork production in Spain, it is necessary to clarify that this country buys a lot of weaned pigs from other European countries, such as Denmark.

The pig production sector is crucial to the Spanish economy, accounting for 17.15% of final agricultural production and 42.63% of final livestock production [13]. Regarding the evolution of the pig population in Spain (Figure 4), it remained stable at around 25–26 million heads during the 2007–2013 period before increasing by over one million heads in 2014 (+4.15%), then seeing a 6.7% increase in 2015. However, during 2022, there was a slight census decline of around 1.1%, recovering in 2023 to reach a total pig population size of 34.452 million animals [13].

When discussing pig production in Spain, it is important to highlight the uniqueness of the coexistence of two major different breed groups. Most of the production is linked to the rearing of commercial pigs based on commercial line crosses (for example, F1 females from Landrace and Large White lines, and Pietrain, Duroc, or hybrid terminal sires), which represent about 90% of the total pig populationhe (Figure 4) [13].

On the other hand, the Iberian pig sector is associated with an indigenous breed from the Iberian Peninsula, the Iberian pig. The production of Iberian pigs accounts for a population of 3,613,123 animals [13], which represents approximately 10% of the total pig census in Spain, whose evolution has remained more stable in recent years (Figure 4). The Iberian pig is known for its high-quality meat and is traditionally associated with extensive systems in the dehesa agoecosystem [14]. However, due to a growing consumer demand for this type of meat, Iberian pig production has largely shifted to intensive systems similar to those used for commercial pigs [15]. As a result, Iberian pigs produced in extensive systems, also known as “acorn-fed” pigs, only represent 17.6% of total production, while Iberian pigs raised in intensive systems account for the majority at 66.9% of production [16].

### 1.3. Common Benchmarks to Evaluate Sow Efficiency

Benchmarking is defined as a systematic and continuous process for comparatively evaluating products, services, and work systems in organizations with the goal of meeting or exceeding defined standards [17]. However, the evolution of this concept over time and in different fields of application explains the multiplicity and heterogeneity of its definitions [18]. In pig production, the most common use of this technique is aimed at organizing and comparing farm performance [19]. Thus, benchmarking as an evaluation and comparison tool allows companies in the pig sector to establish targets, identify best practices, learn from other producers, design competitive strategies, and implement improvement plans to enhance efficiency and achieve better production outcomes [20].

The use, analysis, and comparison of data to improve efficiency and performance in pig production is a very useful tool that has been employed for decades [21,22,23]. Over the years, the number of published studies seeking to explore and take advantage of the large amount of data generated by this type of livestock production has steadily increased (Figure 5) [24,25,26,27,28,29].

In these comparisons, it is important to understand the distribution of results within the dataset: the performance of average farms, as well as those showing the best and worst parameters, using the upper or lower 10th or 25th percentiles of measured performances as the target value [20]. This allows for setting objectives based on what others have achieved; establishing intervention levels indicating that things are not going well on the farm in order to trigger alerts; understanding a farm’s position relative to others; and making comparisons of performance evolution over time [30].

Traditionally, a set of production parameters which can be classified as classic or common at the farm level have been evaluated and compared [31]. These parameters allow for assessing the productive efficiency of a sow farm: annual productivity, measured as the total number of weaned piglets per sow per year (PWSY) [32,33]; farrowing rate, calculated as the percentage of matings that result in farrowing [34,35]; weaning-to-estrus interval (WEI), which corresponds to the period between the day of weaning and the first day the sows show standing heat [36,37]; pre-weaning mortality (PWM), which refers to the percentage of piglets that die during lactation [38,39]; and sow prolificacy, measured as the number of total born piglets (TB) and born alive piglets (BA) per litter [29,40].

A useful tool for evaluating these production parameters at the farm level is the productivity decision tree, which links different production variables [41] to assess farm efficiency, reflected in the total number of PWSY as a reference measure for comparing farm productivity. Thus, Koketsu et al. [31] suggested that it is possible to reach 40 PWSY by meeting the objectives parameters set in the productivity decision tree (Figure 6).

In this context, it is important to study how sow productivity and efficiency have evolved over time. The aim of this study is to analyze these changes, providing an overview of historical trends in sow productivity. This analysis is based on data collected from the BDporc^®^ [42] database, which offers a detailed dataset of the Spanish pig sector as well as additional data provided by InterPIG [11] and previous studies at the international level. Additionally, the study seeks to propose new directions for future evaluations of sow efficiency in light of emerging challenges.

## 2. Materials and Methods

This study analyzed the evolution of sow productivity and the evaluation of sow efficiency parameters based on data from InterPIG (an international network that compares production costs and productive efficiency in the swine industry), previous studies, and data collected from the BDporc^®^ database which provide a comprehensive dataset of the Spanish pig sector. BDporc^®^ has been compiling data about commercial pig farms since the 1990s, and, since 2015, it also has been incorporating information specific to the Iberian pig sector. For this research, a collaboration agreement between the University of Córdoba and the Institute of Agrifood Research and Technology (IRTA) facilitated access to the BDporc^®^ database. This dataset included data from between 525 and 623 Spanish commercial sow-breeding farms, depending on the year, with a total census of up to 883,783 commercial sows (primarily hybrids of Landrace and Large White crosses), and between 35 and 83 Iberian sow farms, with a total census of up to 66,934 Iberian sows (Table 2).

### 2.1. Evaluation of Common Benchmarks

Traditional production benchmarks, as described in the scientific literature, were analyzed both at the global and European Union levels. This was achieved through a comprehensive literature review, selecting relevant studies from leading scientific databases, such as Web of Science and Scopus, focusing on pig production, benchmarking, and sow productivity. From this information, the average evolution of common benchmarks has been presented. Additionally, data from InterPIG were used to study the evolution of annual productivity (PWSY) in the main producing countries.

Furthermore, Spain was selected as a benchmarking reference, being the main pig producer in the EU [13]. To analyze the evolution of farm productivity in Spain, sow performance data from the BDporc^®^ database from the past 10 years were evaluated, covering both commercial and Iberian sows and representing approximately 40% of the sow-breeding census in Spain. This dataset comprised data from 2014 to 2023 coming from commercial and Iberian Spanish farms (Table 2). The key productivity indicators evaluated included the number of weaned piglets per litter, annual productivity (PWSY), piglet mortality rates, and prolificacy trends (piglets BA). These parameters were chosen due to their relevance in evaluating sow efficiency and overall farm productivity [31].

### 2.2. Evaluation of New Benchmarks to Assess Sow Productivity

In addition to the analysis of traditional production benchmarks, the study also explores new indicators that could be integrated into the evaluation of sow efficiency. These emerging benchmarks aim to provide a more holistic approach to productivity assessment, incorporating sustainability, animal welfare, and reproductive performance. This was achieved through a comprehensive literature review that included a search for new evaluation parameters. Relevant studies were selected from leading scientific databases, such as Web of Science and Scopus, focusing on recent research in the areas of genetic improvement, management practices, and the environmental impacts of the pig industry.

Finally, by combining the BDporc^®^ data analysis and bibliographic research, the study offers an overview of Spanish historical trends in sow productivity while also proposing new directions for future efficiency evaluations in the context of emerging challenges, such as reducing the carbon footprint, improving biosecurity, and enhancing animal welfare.

## 3. Results and Discussion

### 3.1. Evolution of Common Benchmarks to Evaluate Sow Productivity

#### 3.1.1. Global Evolution

Advances in sow reproduction have had profound effects on the structure, production, efficiency, and profitability of the pig industry over time [43].

From the 1960s to the 1990s, significant advancements in the productivity of commercial sows were achieved, largely due to improved nutrition, management, and genetic selection [44]. According to data published in the UK’s Meat and Livestock Commission (MLC) yearbooks from 1970 to 2003 [45], the number of litters per sow per year improved rapidly during the 1970s (Figure 7a) due to better management of mating and artificial insemination and the improved detection of sow estrus, which reduced the farrowing interval [44]. However, little improvement was seen in this index from 1980 to 2000, maintaining around 2.2 litters per sow per year during that period [45]. In the last decade, Weaver et al. [46] reported production levels of 2.4 litters per sow per year, which have remained steady in commercial pig farms [47,48].

Regarding the average number of BA piglets per litter, this remained stable from 1970 to 1982, at around 10.2–10.4 piglets. From that decade onward, English et al. [22] proposed the first production goals and measures to improve annual farm productivity, setting a target of 25.8 PWSY. When compared to the average of 20.1 PWSY of that time, this represented a potentially achievable goal of +5.7 PWSY. Beginning in the 1990s, genetic selection programs implemented by sow-breeding companies led to higher prolificacy levels [44], reaching an average of >11 BA piglets from the year 2000 onward. For example, in France, there has been an average annual increase of 0.17–0.23 BA piglets per litter since 1990 [49]. Similarly, in Spain, the average prolificacy has yearly increased by 0.20 BA piglets per litter each year in commercial pig farms since 1990 [50], with the most significant rise occurring after 2010 (Figure 7b).

In terms of PWSY, production performance in the EU, the USA, and Canada from 2010 to 2022 (Figure 8) shows an upward trend, with an average annual increase of +0.48, +0.29, and +0.30 PWSY, respectively [11]. In Europe, this increase varies between countries, with Belgium being the leader with an annual increase of +0.54 PWSY, followed by Germany with +0.53 PWSY. In contrast, France (+0.31) and Spain (+0.27) have shown moderate growths during this period. Denmark, due to its already high productivity in 2017, with a total of 33.29 PWSY, remained stable in recent years, reaching a maximum of 34.14 PWSY in 2022. In Brazil, InterPIG provides data for two key regions in the country’s pig production: Mato Grosso and Santa Catarina, which have both showed an upward trend since 2010, with an average annual increase of +0.28 and +0.46 PWSY, respectively, and an average annual productivity in 2022 of 28.42 and 29.71 PWSY, respectively (Figure 8) [11]. In contrast, China currently shows an average annual productivity of only about 20 PWSY [51,52] despite being the world’s largest pork producer. However, studying the evolution of sow efficiency in China is difficult, as scarce global data are available for the country.

This PWSY production increase is largely due to the improvements achieved through years of genetic selection of hyperprolific lines, supported by the widespread use of Best Linear Unbiased Prediction (BLUP) methodologies and advances in genomic selection, both of which have significantly enhanced the accuracy of breeding programs [53,54,55]. These advancements, along with the integration of management software, have allowed commercial farms to optimize decision-making processes, enhancing overall productivity. Hyperprolific sows are considered those whose litters consist of 16 or more TB piglets [56] or those that give birth to more piglets than they have functional teats [57]. However, this increase in prolificacy can negatively affect piglet mortality during lactation [58]. Thus, PWM on European farms, which was at around 12% in the first decade of the 21st century, representing a loss of approximately 39 million BA piglets per year [59], now ranges between 15% and 20% [60]. However, in the USA there is a slightly lower PWM average of around 14% [39].

To address this sustainability challenge, genetics companies are focusing more on producing viable piglets (piglets with higher birth weights, which have a greater chance of surviving during lactation) [61] rather than continuing to increase the number of piglets BA, aiming to reduce PWM. In this sense, Denmark, which currently has the most prolific lines, has maintained relatively constant productivity over the last 5 years (+0.17 PWSY yearly) in an effort to reduce its PWM [10], which followed the increase in PWSY.

Thus, in this global evolution of productivity, the achievements in prolificacy and the ongoing challenges faced by the pig industry to reduce piglet mortality stand out, highlighting the need to continue developing more sustainable and effective management practices.

#### 3.1.2. Evolution in Spain

In Spain, numerous studies analyze the productive performance of commercial pigs based on commercial lines, especially reproductive sows, focusing on their reproductive efficiency [24,25,26,29,62,63,64]. Similarly, an increasing number of studies also evaluate the reproductive efficiency of Iberian sows [27,65,66,67,68], leveraging the substantial amount of data generated by the high level of intensification and technology achieved in recent years.

Currently, thanks to information provided by BDporc^®^, a database that has focused on the pig sector since the 1990s, and which has incorporated data from the Iberian pig sector since 2015, the evolution of the main production parameters for both commercial pigs and Iberian pigs in Spain is well known [42]. This information is highly useful for setting goals and reference levels for the sector.

Regarding the classic production parameters, the latest data available from 2023 show that the average annual productivity of Spanish commercial sows is 29.45 PWSY (Table 3), which is lower than the EU average productivity of 30.16 PWSY. This EU average is elevated by the high productivity of countries such as Denmark (34.14), The Netherlands (32.47), and Belgium (31.36) [11]. However, Spain’s average productivity is higher than that of other competitor countries such as the USA or Brazil [11]. In contrast, the current average annual productivity of Iberian sows is 17.44 PWSY (Table 3), which reflects significant differences between the two racial groups [69].

Thus, the productivity of Spanish commercial sow farms has increased, although the figures remain slightly lower than in the rest of Europe, rising from 23.78 PWSY in 2009 [70] to the current 29.45 PWSY [42]. Naturally, annual productivity is the result of improving production parameters such as prolificacy and the number of piglets weaned per litter with an accompanying increase in PWM over the past several years in Spain [26].

As for the Iberian pig sector, few studies have examined the productivity of this breed in the past, and those that exist show considerable variability in their results. Nieto et al. [15] compiled a total of 19 studies conducted between 1983 and 2013, indicating an average prolificacy of 7.5 piglets BA per litter for Iberian sows, although data on other parameters determining productivity are scarce. Faced with this lack of knowledge regarding Iberian sow productivity, Piñeiro et al. [71] highlighted the need to establish reference levels for Iberian sows, which in 2009 had a prolificacy of 8.3 TB and 7.8 BA per litter, with an average annual productivity of 14.4 PWSY. Nowadays, further studies focused on the productivity of this racial group are still needed, given its importance in the pig sector within the leading pork-producing country in Europe.

Figure 9 shows the evolution of prolificacy (piglets BA per litter) and piglets weaned per litter on average from commercial and Iberian farms in Spain over the past 10 years [42]. In both cases, the increase in prolificacy and piglets weaned per litter is noteworthy, reaching average figures of 15 and 8.3 piglets BA, and 12.3 and 7.3 piglets for commercial and Iberian sows, respectively, in 2023. This represented an average increase over the last 10 years of +2.27 and +0.62 piglets BA, and +1.12 and +0.34 weaned piglets in commercial and Iberian farms, respectively. However, it should be noted that this increase in prolificacy has been accompanied by a rise in piglet mortality during lactation, reaching up to 18.21% and 12.82% PWM in 2023 on commercial and Iberian farms, respectively. These results translate into an average annual increase of 0.65% and 0.29% in PWM since 2014 in commercial and Iberian farms, respectively. This is a concern for the sector and also presents a challenge in terms of animal welfare and the sustainability of farms [72,73].

In 2021, Spain achieved its highest historical annual productivity, with an average of 30.3 and 17.6 PWSY for commercial and Iberian sow farms, representing an increase of +2.7 and +1.1 PWSY since 2014, respectively (Figure 10). These unprecedented results are mainly due to the genetic improvement of the lines used as well as improvements in the reproductive management of both commercial [64] and Iberian sows [67].

However, in 2022 and 2023, this annual and progressive increase in annual productivity slowed (Figure 10), with commercial sow farms showing an average below 30 PWSY, returning to productivity levels similar to Spain’s 2018 average [69]. This decline in productivity could be explained by severe outbreaks of a highly virulent strain of the porcine reproductive and respiratory syndrome virus (PRRSV-1) known as “Rosalía” which was first recorded in northwest Spain in 2020 and spread to the rest of the country during 2021 and 2022 [74,75]. These PRRSV outbreaks are associated with a significant increase in abortion rates, stillborn piglets, and piglet mortality during lactation, which explains the poorer figures recorded in recent years [76]. However, it is expected that these indices will continue to improve in the future, following the upward trend of pig production in Spain over the past decade [26].

### 3.2. Other Benchmarks to Evaluate Sow Productivity and Efficiency

The PWSY benchmark encompasses most of the classic production parameters (litters per sow per year, prolificacy, WEI, piglets weaned, etc.). However, it does not reflect some shortcomings (stillborn piglets, PWM, second litter syndrome, early sow removal, etc.). Moreover, the study and evaluation of classic production parameters requires an in-depth analysis of other underlying parameters and factors that affect these main indices. This is especially useful in a sector that is constantly evolving and features increasing demands for greater sustainability [77].

Therefore, in addition to the previously mentioned classic production parameters, there are many other parameters that show farm efficiency and have been studied to set production goals at the farm level, as described below.

#### 3.2.1. Non-Productive Days and Inefficient Days per Sow

Non-productive days per sow (NPDs) are the days when a sow or gilt of reproductive age is neither pregnant nor lactating [23] and which are associated with farrowing rate, WEI, and the weaning-fertile-estrus interval, among other elements [78]. These NPDs represent an issue in terms of farm efficiency and economic sustainability [79], and currently average around 13 days per reproductive cycle [78]. On the other hand,herd-life non-productive days average between 72 and 77 days in commercial sow-breeding farms, which encompasses the period from the weaning-to-first-mating interval and any re-service intervals occurring from the sow’s initial mating as a gilt until its removal during herd-life [25,80]. To reduce these NPDs, Koketsu et al. [25] show that high-productivity herds (>26.5 piglets) have an average of 66 herd-life non-productive days, which could be proposed as a benchmark goal for other herds. According to Rodríguez-Estevez and Perez-Marín [81], in commercial farms, only 2.7% of sows would come into estrus before the 4th day post-weaning, while 26.4% would come into estrus on the 4th day and 41.8% would come into estrus on the 5th day post-weaning, respectively. Therefore, the three NPDs after weaning, prior to coming into estrus, could be considered physiological NPDs that the sow requires to come into estrus. Therefore, De Andrés et al. [82] propose introducing a new concept: “inefficient days”. These would correspond to traditional NPDs but without counting unavoidable intervals like the WEI. Therefore, “inefficient days” would refer to those days in which a sow does not produce piglets due to some reproductive inefficiency, such as returns to service, among others. In this way, this concept provides a more precise indicator for analyzing and evaluating reproductive issues on a farm.

#### 3.2.2. Pre-Weaning Piglet Survival Rates and Piglet Livability

Among the concerns in the production sector related to animal welfare, beyond compliance with welfare regulations and standards, is the decline in piglet survival rates. As mentioned in previous sections, mortality during lactation has become a major issue in modern intensive swine production [59], being largely associated with the increased prolificacy of hyperprolific sows [83] and the piglet survival rates during lactation, which range between 80 and 85% in commercial farms [60]. Therefore, improving piglet survival must be included within the concerns for animal welfare and farm sustainability [72], studying piglet survival during birth and lactation [84] and setting reference targets to improve current survival outcomes. Sanz-Fernández et al. [28] propose that commercial sow farms aim for a minimum of 88.5% piglet survival during lactation and 83.2% survival when including piglets TB. Many authors propose strategies to improve these survival rates, aiming to reduce stillborn and pre-weaning deaths [61,85,86,87]. In this context, another productive parameter known as “piglet livability” has emerged, which refers to the percentage of potentially viable piglets a sow is capable of raising [88]. In addition, piglets with low birth weights have an increased risk of mortality during lactation [89]; therefore, birth weight can also be considered a key indicator or parameter for assessing a piglet’s likelihood of survival [61]. This is currently used by many genetics companies to select their sow-breeding lines; however, sometimes it is a hard metric to record under commercial conditions.

#### 3.2.3. Sow Longevity and Lifetime Performance

Another social concern regarding animal welfare on sow farms is the reduction in sow longevity and premature culling [90]. Culling and mortality in breeding herds from commercial farms have steadily increased over the past 10 years, especially among young sows [91]. Similarly, in terms of longevity, Spanish commercial sows were culled after an average of 3.9 farrowings in 2023, with an average reduction of −0.63 farrowings per culled sow over the past 10 years [42] (Figure 11). A replacement rate of around 40–50% is currently considered appropriate [92]. These results reflect early sow removal, leading to an increase in the replacement rate. Therefore, this is not just an animal welfare issue but also a sustainability and profitability problem, as lower sow amortization increases production costs and the system’s carbon footprint [93]. Many authors suggest improving sow management from birth and throughout their lifetime to increase longevity, improve animal welfare, and reduce farm costs [94,95]. Another parameter that can be included to evaluate farm productivity and efficiency, and which is related to sow longevity, is the total number of piglets weaned per culled sow. This can be considered the sow’s lifetime production and will depend on the annual productivity (PWSY) and the total number of farrowings per culled sow, among other factors [96]. In these terms, the total number of piglets weaned per culled sow in Spanish commercial farms has decreased from 52.61 piglets in 2020 to 47.34 piglets per culled sow in 2023, falling below the 2014 results (Figure 11).

Lucia et al. [97] indicated that the most common reasons for culling sows were related to reproductive issues (33.6%) and suboptimal litter performance (20.6%), with 45% of culls occurring among gilts or sows that had only farrowed once or twice. Therefore, studying and evaluating early farrowings is essential for determining and predicting sow performance throughout their life [98]. Several authors suggest that sows with higher prolificacy in their first two farrowings continue to be the most productive in the future and have longer lifetimes [99,100]. To ensure greater future performance and longevity, it is crucial to identify possible reproductive issues in these early cycles, such as second litter syndrome, which involves a reduction or maintenance of sow productivity in the second cycle compared to the first [101] and may not be detected at the farm level. Several studies conducted on different breeds in the United States, Japan, Mexico, Brazil, Spain, and The Netherlands showed that the incidence of this syndrome ranged from 33.3% to 56.6% [27,98,101,102,103,104,105,106,107]. Therefore, the percentage of sows affected by this syndrome should be considered a benchmark. Improving sow management and identifying factors to reduce this syndrome are key strategies for enhancing farm efficiency.

#### 3.2.4. Herd Age Structure

Sow longevity and replacement policies will determine the age structure or population structure of the herd, which will, in turn, affect reproductive efficiency. When organizing a farm, it is essential to study the age distribution of the breeding sow population to maintain a stable and productive herd structure with an appropriate replacement and culling policy [108]. Vizcaíno et al. [92] suggest minimizing sow loss in early cycles, extending productive life, and achieving an optimal age distribution to ensure good productivity and sow longevity. In this regard, Mote et al. [109] indicated that sow losses should be limited to no more than 10% between one cycle and the next to maintain an adequate herd structure, while Buxadé et al. [110] suggested maintaining a higher percentage of sows up to the third or fourth parity to maximize productive performance and ensure sow amortization. Traditionally, Carroll [111] defined an ideal herd structure as one where the percentage of sows decreases steadily as the number of parities increases; however, more recent authors propose maintaining a higher percentage of sows in intermediate parities (third to fifth), allowing them to reach their maximum reproductive potential, reducing the necessary replacement rate, and increasing the productive life of sows [48,112]. Hence, the herd age structure should be analyzed as a benchmark.

### 3.3. Emerging Challenges and Future Directions for the Pig Production Evaluation

In pig production systems, one of the main critical points that must be addressed is pig health. Health is a core component of animal welfare and reflects the optimal state of the animals, which directly impacts greater productive performance and farm efficiency [113]. Therefore, constant monitoring of production parameters can also be a useful practice, allowing for early detection of health issues [114]. Precision livestock farming, which uses advanced technologies and the analysis of data generated by farms, provides tools for real-time monitoring the animals’ health [114]. These systems can collect and analyze large volumes of data, identifying patterns and anomalies that may indicate health problems. Implementing these systems allows for informed decision-making to improve health, increase productivity, and reduce animal mortality [115].

Other key challenges include sustainability and environmental impacts of swine production. In addition to the global challenge of minimizing the environmental impact of animal production, consumers are demanding products with a lower carbon footprint [116]. In Mediterranean climate zones, swine production is estimated to account for 11% of emissions related to livestock farming [117], taking into account the environmental footprint derived from feed production, the animals themselves, and their waste [118]. Beyond feed, the efficiency of which can be improved with precision feeding systems [119], the reduction in production efficiency often leads to inefficient resource use and, therefore, greater impacts per unit of meat produced [120]. In this sense, Rolim Pietramale et al. [121] pointed out that sow reproductive problems impact the carbon footprint of swine production, showing a lower carbon footprint in farms with higher PWSY levels, greater efficiency, and fewer NPDs. Similarly, Tallaksen et al. [116] indicated that high-productivity farms consumed 11% less energy than intermediate-productivity farms and 25% less than low-productivity farms. Therefore, from a sustainability standpoint, there is also a need for more efficient animal management to improve sow productivity and fertility, reduce early culling, and lower piglet mortality to reduce the environmental footprint [122].

Finally, another major social demand is the responsible use of antibiotics, recognizing that, although they are essential for treating certain infections in animals, their use must be regulated to avoid contributing to antimicrobial resistance, a problem that affects both animal production and public health [123]. Therefore, it is important to continue working on the development of tools and evaluation methodologies for farm management and hygiene to reduce the use of medication and improve farm efficiency, including aspects related to biosecurity, production flow, medication use, and animal welfare [124,125,126,127]. Moreover, studies evaluating measures to reduce the use of antimicrobials, as well as indices assessing farm management and hygiene, are essential to improve animal welfare [124,128,129,130,131]. Promoting the responsible use of antibiotics is essential for the health of animals, people, and the environment, which are closely interconnected, as established by the One Health principle [132]. Thus, the classic decision trees of the swine industry [31,41] should be updated to include parameters related to health or animal welfare [133], which is extremely useful in a sector that is continuously evolving.

## 4. Conclusions

Continued efforts to collect and analyze the above-mentioned new parameters are essential to establish a solid database for accurately evaluating the evolution of these benchmarks over time. By incorporating these indicators, farms will be better prepared to face the current and future challenges of the swine industry while ensuring the sector’s competitiveness and sustainability.

Overall, benchmarking is a powerful tool for evaluating and improving the efficiency of sow herds. By comparing key performance metrics and setting improvement targets, producers and veterinarians can identify problem areas and develop action plans to enhance productivity and herd health. Establishing new reference parameters based on real data will enable the creation of new strategies to deepen the evaluation of farms, especially with growing demands for greater sustainability. Thus, the swine industry must adapt to future challenges by adopting new parameters that consider not only productive efficiency but also animal welfare, sustainability, and health. These advances will contribute to a more responsible, efficient, and resilient swine industry that is capable of meeting the expectations of consumers and society as a whole, allowing producers and technicians to compare their results and continue improving.

## Figures and Tables

**Figure 1 vetsci-11-00626-f001:**
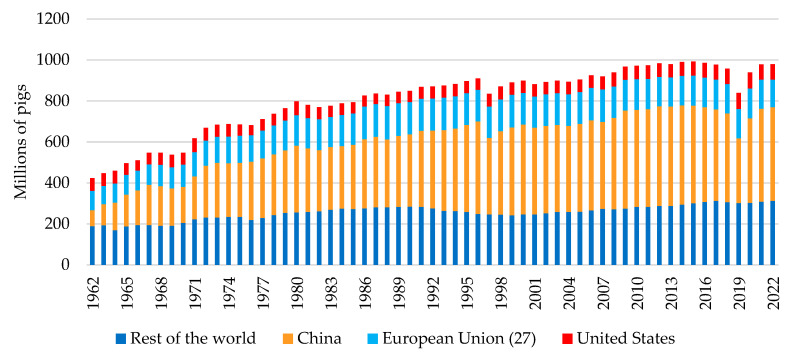
Global number of pigs produced in the main producing countries from 1962 to 2022 [3].

**Figure 2 vetsci-11-00626-f002:**
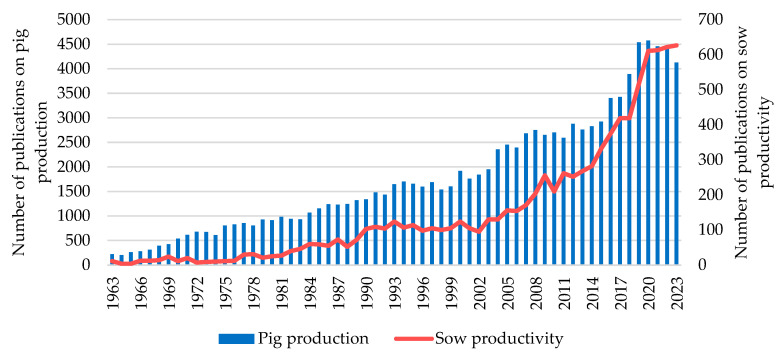
Number of total publications related to pig production and sow productivity from 1963 to 2023 in the “Web of Science” database [6] (terms of search: “pig production” and “sow productivity”).

**Figure 3 vetsci-11-00626-f003:**
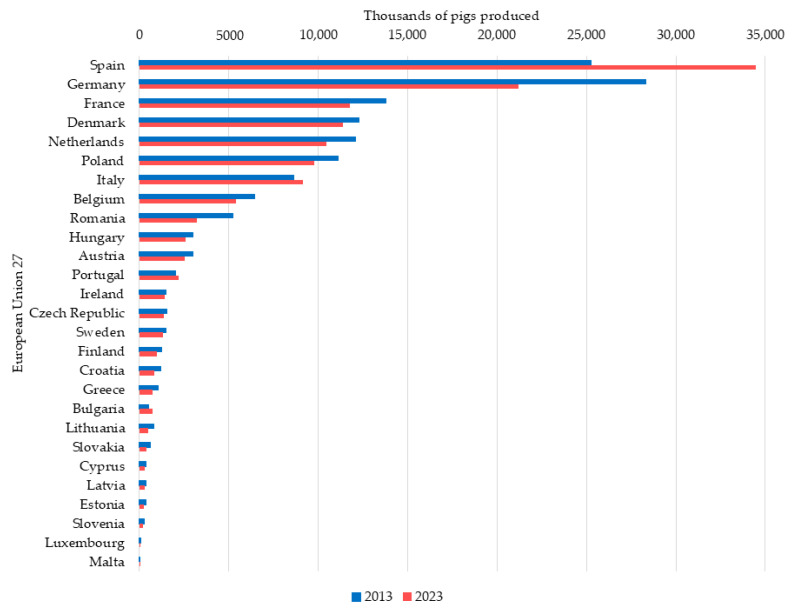
Evolution of the pig census among the top pig-producing countries in the European Union 27 over 10 years (2013–2023) [12].

**Figure 4 vetsci-11-00626-f004:**
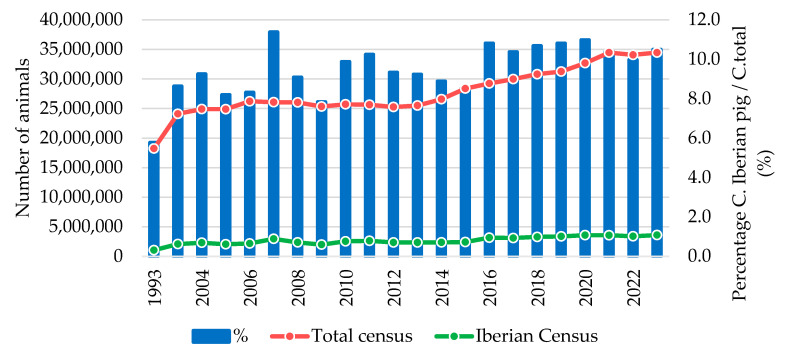
Evolution of the total pig census and the Iberian pig census in Spain (1993 to 2023) [13].

**Figure 5 vetsci-11-00626-f005:**
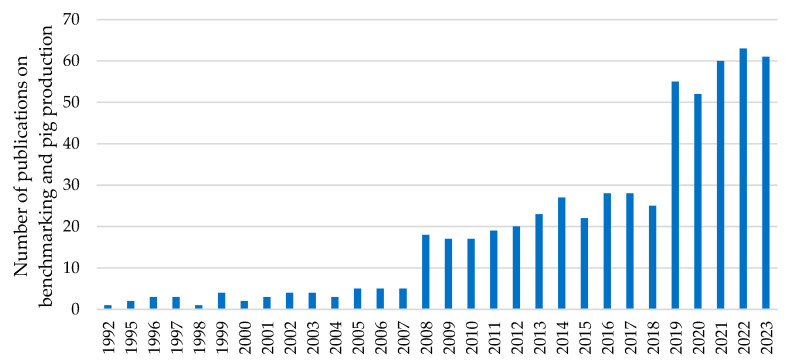
Total number of publications related to benchmarking and pig production from 1992 to 2023 in the “Web of Science” database (terms of search: “Benchmarking and pig production”).

**Figure 6 vetsci-11-00626-f006:**
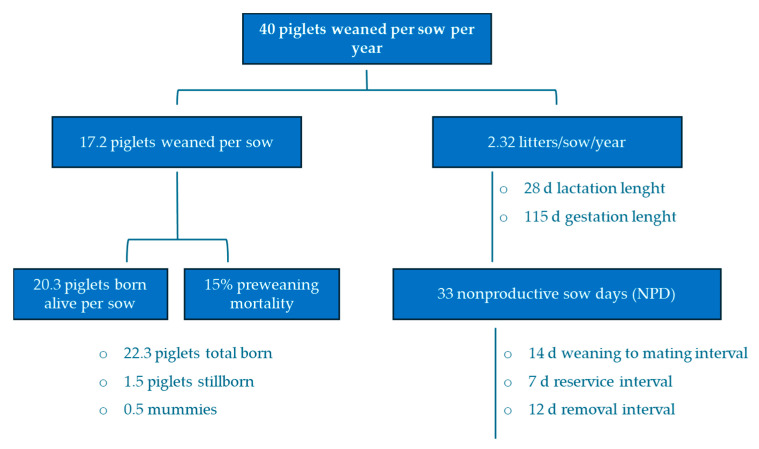
Productivity decision tree of a farm for analyzing the evolution of classic production parameters with the goal of reaching 40 PWSY. Adapted from Koketsu et al. [31].

**Figure 7 vetsci-11-00626-f007:**
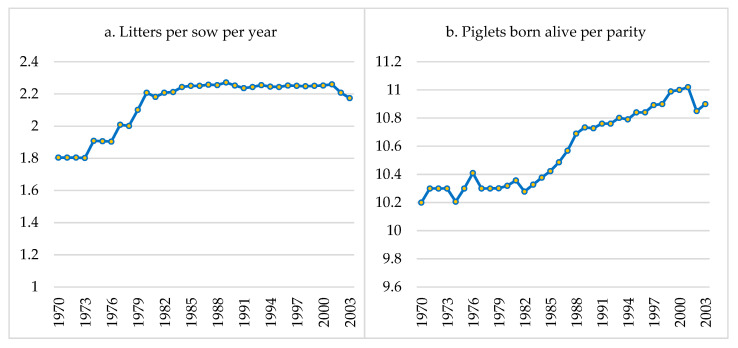
Evolution of litters per sow per year (**a**) and number of piglets born alive per litter (**b**) from 1970 to 2003. Adapted from the MLC [45].

**Figure 8 vetsci-11-00626-f008:**
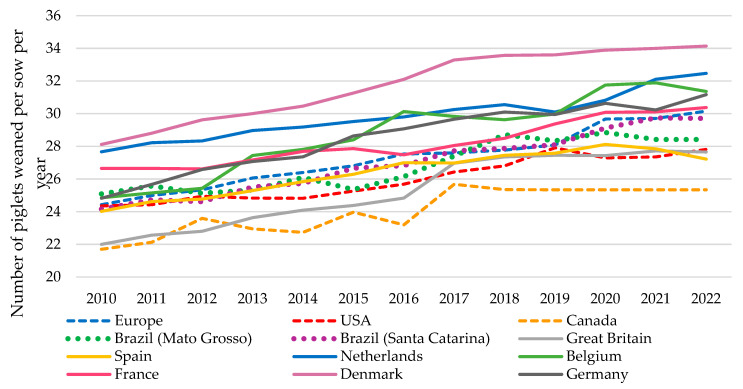
Evolution of weaned piglets per sow per year in selected countries [11] (note that this graphic comes from an InterPIG report which contains a small number of representative farms; therefore, these figures are not exactly accurate).

**Figure 9 vetsci-11-00626-f009:**
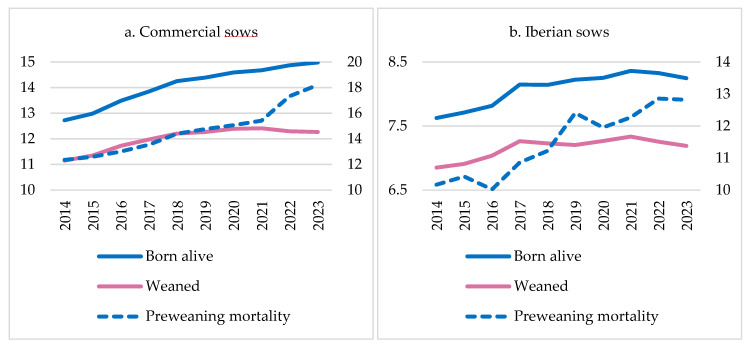
Evolution of piglets born alive and weaned from (**a**) commercial sow and (**b**) Iberian sow litters in Spain from 2014 to 2023.

**Figure 10 vetsci-11-00626-f010:**
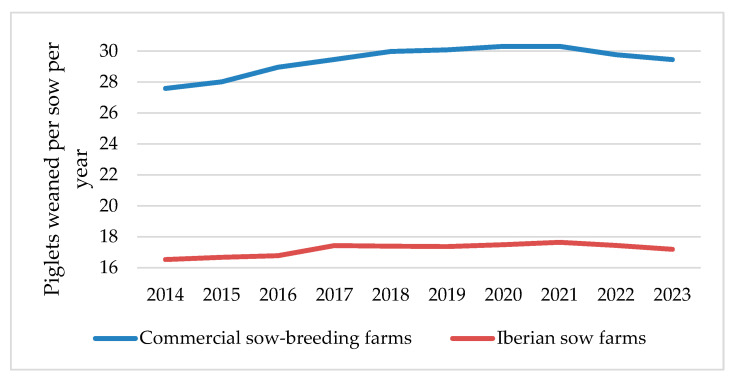
Evolution of annual productivity of commercial and Iberian sow farms from 2014 to 2023.

**Figure 11 vetsci-11-00626-f011:**
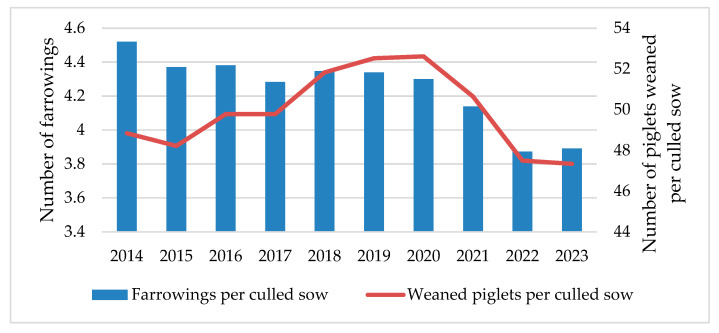
Evolution of farrowings and piglets weaned per culled sow in Spanish commercial sow farms from 2014 to 2023.

**Table 1 vetsci-11-00626-t001:** Pork production in the world’s main producing countries [13].

Producing Countries (Million Tons)	2022	2021	2020	2019	2018	2015	2012
China	51.00	47.50	36.34	42.55	54.04	54.87	53.43
USA	12.32	12.56	12.85	12.54	11.94	11.12	10.55
Spain	5.07	5.18	5.00	4.64	4.53	3.86	3.47
Brazil	5.05	4.37	4.13	3.98	3.76	3.52	3.33
Germany	4.49	4.97	5.11	5.23	5.34	6.21	5.2
Russia	3.80	3.70	3.61	3.32	3.16	2.62	2.18
Vietnam	2.75	2.59	2.47	2.43	2.81	2.48	2.31
France	2.15	2.21	2.20	2.20	2.18	1.89	2.23
Canada	2.06	2.12	2.12	2.00	1.96	1.90	1.84
Poland	1.79	1.98	1.97	1.98	2.08	2.11	1.77
Netherlands	1.71	1.72	1.66	1.63	1.54	1.5	1.56
Denmark	1.63	1.72	1.6	1.5	1.58	1.56	1.64
Mexico	1.53	1.48	1.45	1.41	1.32	1.32	1.24
South Korea	1.41	1.41	1.4	1.36	1.33	1.22	1.09
Japan	1.31	1.32	1.31	1.28	1.28	1.25	1.30
Italy	1.27	1.34	1.27	1.45	1.47	1.43	1.60
Philippines	1.10	1.00	1.12	1.59	1.60	1.37	1.60
Belgium	1.04	1.14	1.10	1.05	1.09	1.12	1.23
World	110.50	107.61	95.76	101.03	111.92	110.38	106.87

**Table 2 vetsci-11-00626-t002:** Total number of farms and mean number of sows included in the study for the evaluation of common benchmarks based on BDporc^®^ data.

	Commercial Sow-Breeding Farms		Iberian Sow Farms	
Year	Number of Commercial Farms Within BDporc^®^	Mean Number of Sows Included per Farm	Number of Iberian Farms Within BDporc^®^	Mean Number of Sows Included per Farm
2014	613	1143	34	699
2015	604	1199	40	746
2016	596	1238	54	779
2017	618	1338	53	783
2018	592	1343	57	831
2019	583	1381	61	829
2020	623	1419	63	787
2021	559	1440	71	822
2022	525	1532	78	800
2023	536	1554	83	807

**Table 3 vetsci-11-00626-t003:** BDporc^®^ results in 2023 for commercial (a) and Iberian (b) sow farms from the reference pig sector database in Spain.

	a. Commercial Sow		b. Iberian Sow	
	Media	C.V.	Media	C.V.
Total farms	536	-	78	-
Number of reproductive sows	1554	81.18	800	73.99
Number of weaned piglets per sow per year	29.45	12.25	17.44	12.03
Piglets total born per litter	16.63	12.07	8.76	8.92
Piglets born alive per litter	15	11.24	8.33	8.7
Piglets stillborn per litter	1.63	33.75	0.44	56.44
Piglets weaned per litter	12.27	11.2	7.26	8.64
Pre-weaning piglet mortality (%)	18.21	32.25	16.78	34.65
Farrowing rate (%)	84.77	6.94	82.50	11.87
Weaning-to-estrus interval	4.86	14.70	6.15	17.71
Weaning-to-successful mating interval	10.9	36.33	12.33	57.94
Farrowing interval (days)	152	6.98	152	6.90

C.V.: Coefficient of variation.

## Data Availability

These data are not deposited in an official repository due to confidentiality constraints. However, a portion of these data may be available upon request from the author Santos Sanz-Fernández (v22safes@uco.es).

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
