# Peer review of "Evolution of Sow Productivity and Evaluation Parameters: Spanish Farms as a Benchmark"

_vetsci, 2024, doi:10.3390/vetsci11120626_

Round 1
Reviewer 1 Report
Comments and Suggestions for Authors
In this manuscript, authors reviewed the sow procductivity in the major pig prodcucing countries all over the world through comparing the key parameters. the paper was well organized and provided useful data for the industry. Particularly, raising a prospect that make a more responsible, efficient, and resilient pig industry to meet the expectation of sconsumes and society in terms of animal welfare, sustainabilty and health.
Author Response
Please, see the attached file.
Thanks

Reviewer 2 Report
Comments and Suggestions for Authors
Overall, the paper is really good! My major concern is after the results of the "evaluation of new benchmarks" session. Up until this point, the paper was flowing very well, but the new benchmark session does not have a lot of benchmarks like the previous session (evaluation of common benchmarks). I encourage the authors just to describe one benchmark and convert the "new benchmark part" as more of a discussion. The way it is now creates an expectation on the readers that they will see a lot of benchmarks for the variables in this "new benchmark" session, but that does not happen.
Lines 55-56: Prices doubles, leading to a moderate increase? I think sentence needs clarification for the readers. Doubled is not a moderate increase. Doubled where?
Line 64: Perhaps missing a sentence explaining how it was possible to increase production with the same number of pigs.
Line 67: Why you mention 3,000 in 2016 if your chart contains data up to 2016?
Line 78: Can you also add the increase in million of pigs? This way matches the context of the paper as you used this metric on Figure 1.
Lines 85-86: I would try to be consistent with one or both metrics across the paper.
Line 88: "which, together, account..."
Line 90: Denmark also has great productivity levels due to great disease control and elimination plans, especially for PRRSV. If they had PRRS endemic in their herds, they probably would not have these numbers.
Lines 91-94: All during the 10 year period? How much of that decrease in production is due to the recent ASF breaks (after the introduction in each country) or started before?
Line 95: followed by Germany and Spain --> You mentioned 54% today from these 3 countries...maybe include the percentage for each one to complete that value...this will be a good introduction to Figure 3.
I think you mentioned the top 3 countries (and I like that), but France is not that different from Denmark, Netherlands, and Poland. Maybe mention the top 2 which will around account for more than ~43% of EU's production? Just an idea....feel free to consider this or not.
One interesting thing to mention would be the importance of Denmark for example (and other similar countries), where they produce a lot of pigs to be sold to other countries in EU as weaned pigs and the worldwide genetic importance.
Line 101: 52 million slaughtered, but what about the value in Figure 3 (a little over 35000 k pigs)? How many from imported outside? I think Spain buys a lot of weaned pigs from Denmark, correct? Maybe it is important to explain this scenario of pork consumption and pork production.
Line 122: "...in the dehesa grassland" or landscape maybe.
Line 126: Maybe mention the type o diet for the Iberian pigs raised in intensive systems?
Line 168: LDCPA?
Line 144: These studies are more on the descriptive statistics from large aggregated databases and trying to investigate trends. What else are people doing with this large swine data besides this? Why/where publications have evolved?
Line 182: "This study analyzed..."
Line 189: 525 or 623? Explain that this variation depends on the year?
Line 199: You explained BDporc well but not InterPIG.
Line 223: such as? Are there any other metrics? Maybe be more clear on the number of new metrics evaluated compared to the standard metrics used on common benchmarks, clarifying the difference.
Line 233-236: You did not mention LFY between lines 206-209. It would be easy to have a table of all metrics for the common and new benchmark maybe.
Line 254-255: This InterPIG sentence should be on the materials & methods section.
Line 255: "In terms of PWSY, the EU, the USA, and Canada..."
Line 260: delete word "more"
Line 261: "...productivity since 2017...."
Lines 281-284: I think this message is wrong or it does not make a lot of sense. If you look at the PWSY charts, US is inferior to EU, and it is probably not due to lower Born Alive but due to higher PWM rates compared to EU. Make sure to compare average of countries under similar conditions. I encourage the authors to breakdown EU by high production efficiency countries that export pigs versus Spain and other that are similar to Spain in terms of disease challenge etc.
Line 290: I would add how the PWM rates followed the +0.17 increase in PWSY over the last 5 years as well.
Line 308-309: Why does this number appear to be higher than what is displayed in Figure 8?
Line 311-312: How Spain have better numbers if the chart (Figure 8) shows lower numbers than USA and Brazil for PWSY?
Table 3: You mentioned PWM between 15-20% for Spain nowadays on lines 286. The value here shows 18.21% for commercial sows. Why not standardize and report only one format across the manuscript?
Figure 9: I love this chart, especially because we can see the clear effect of disease as mentioned below. I'd like to see that for other countries such as Denmark, or maybe two clusters of EU countries: countries with low disease (PRRS and/or ASF) challenge and countries without disease challenge.
Figure 10: Formatting is split between two pages
Figure 10: I don't think this figure is very important as this information can be seen easily on Figure 8.
Line 376: WEI?
Explain the difference for readers between NDP of 13 days and Herd NDP of 72-77.
Line 395-396: Be careful saying that a high hyperprolific sow will have high PWM. How did Brazil and Denmark avoid that? There is also the disease aspect, labor: sow proportion etc.
Line 409: mention that it is a hard metric to record under commercial conditions, not like by genetic companies.
Line 414: Mentioning the -0.63 reduction was the first time in this section, "3.2 New benchmark," that the authors actually state some benchmarking information; all the previous ones were descriptive information or snapshots of the current scenario for the variables.
Author Response
Please, see the attached file.
Thanks
